# Coevolutionary dynamics in the grass-livestock social-ecological system of China's alpine pastoral areas: A case study of the Qilian Mountains region in China

Ligan Cai[1], Junhao Zhao[2], Jian Chen [3]*

**1** Soil and Water Conservation Work Station, Subei County Agriculture, Rural Affairs and Water Affairs Bureau, Subei Mongol Autonomous County, Gansu, China, **2** College of Veterinary Medicine, Gansu Agricultural University, Lanzhou, Gansu, China, **3** Department of Geography, University College London, London, England, United Kingdom

\* jian.chen.21@ucl.ac.uk

## Abstract

Evaluating the dynamic co-evolution and feedback mechanisms within socio-ecological systems is crucial for determining the resilience and sustainability of environmental governance strategies. The grass-livestock system, as a complex entity encompassing livestock nutrition, foraging behavior, vegetation ecology, pastoralists' economic income, and policy interventions, indicates that any change in a single element may trigger a chain reaction within the system. This paper uses a system dynamics approach to construct a simulation model of the grass-livestock system in alpine pastoral areas, simulating the long-term dynamic co-evolution of the socio-ecological system in the Qilian Mountains region of China. Four optimization schemes were proposed, and the synergistic development of the grass-livestock system in alpine pastoral areas under each scheme was simulated. The results show that, under the premise of sustainable use of grazing-based artificial grassland, the combination of reasonable use of fenced grasslands and cooperative management by pastoralists can effectively control livestock numbers, ensure pastoralists' income, and maintain grassland quality within the next 20 years, thereby achieving coordinated socio-economic and ecological development. Additionally, optimizing feed supply can significantly improve grass production, livestock weight, and income. Therefore, it is recommended that alpine pastoral areas prioritize both grassland ecological management and development, adopt grassland restoration technologies, strengthen the management of artificial grasslands, set reasonable grazing bans, develop pastoralist cooperative organizations and design internal operational mechanisms.

## 1. Introduction

China's 400 million hectares of pastoral land rank as the world's second-largest, covering approximately 40% of the country. These lands are located in the drier, higher regions of north and northwest China and are primarily inhabited by various ethnic minorities [1]. China's alpine pastoral regions nurture numerous mountains, rivers, and streams, significantly

**Data availability statement:** All relevant data are within the manuscript and its Supporting Information files.

**Funding:** The author(s) received no specific funding for this work.

**Competing interests:** The authors have declared that no competing interests exist.

influencing the ecology of the northwest region and even the globe, while providing essential material support for local production, living, and livestock development [2–4]. Therefore, their ecological and production functions are crucial for China's biodiversity, the social stability of pastoral areas, and the survival and development of pastoralists [5,6].

The transformation of China's alpine pastoral areas is currently facing many challenges, especially due to the irrational number of grazing animals [7,8] and the imperfect system of grassland resource management [9], the alpine pastoral areas are faced with the problem of the continuous deterioration of grassland ecology [10]. Over the past 20 years, 90% of China's usable grasslands have experienced varying degrees of degradation, with about 42% of the grasslands in alpine pastoral regions degraded [11]. Without effective measures, grassland resources will continue to deteriorate, necessitating the strengthening of grassland ecological construction and protection. Facing the dual pressures of ecological degradation and pastoralists' livelihoods [12], The State Council, the Ministry of Agriculture and Rural Affairs, the State Forestry and Grassland Administration of the People's Republic of China (PRC) have gradually built up a grassland management system based on the grassland contract responsibility system [13], the grassland grazing ban, the construction of artificial grassland and other major projects, laws and regulations and policy measures have been implemented [14]. However, it remains to be seen whether the current management system in alpine pastoral regions has achieved the expected results and meets the production and living needs of pastoral areas [15]. In practice, promoting the ecological protection of grasslands and the socio-economic development of pastoral areas continues to encounter issues of imbalance, incoherence, and unsustainability [16–18]. Exploring models and systems for the coordinated development of grassland ecology and livestock economy has become a significant task and challenge for the transformation and development of China's alpine pastoral regions [16,19].

In the mid-to-late 1990s, China implemented a new round of grassland property rights reform. The pastoral areas widely began implementing the grassland contracting responsibility system, transferring the use rights of grasslands to households [14,15,18]. Pastoralists marked the boundaries of their contracted grasslands with fences, clarifying their usage rights. This policy, which aims to effectively address the stagnation of pastoral development by defining property rights, is thought to have greatly mobilized pastoralists' livestock production [20–22]. However, the implementation of this policy led to the 'fence trap' in grassland management. The fragmentation of grasslands limits livestock mobility and disrupts the connections between grassland ecosystems. Pastoralists, constrained by the scale and capacity of their grasslands [23], find it difficult to implement reasonable grazing arrangements based on climate changes and grassland quality [24,25]. Continuous grazing in the same area over the long term hinders the natural recovery of grasslands, leading to ecological degradation [26]. Furthermore, some pastoralists, as the main production units striving for maximum output, face rising per capita production costs while inevitably increasing the intensity of grassland resource utilization. This results in severe overgrazing and even the overall degradation of grasslands [27,28].

After the implementation of the grassland contracting responsibility system, the problem of grassland degradation in pastoral areas remains widespread [29]. This policy has faced criticism and controversy from various sectors of society [14,15,30,31]. Some viewpoints suggest that the policy needs further improvement and that a comprehensive grassland management and utilization mechanism should be established based on property rights [32,33]. Leisher (2011) [34], Pinheiro (2021) [35] and Marinheiro (2023) [36] have proposed that collective governance of grassland resources is essential to ensure the sustainable development of pastoral areas. Chinese scholars such as Qi (2023) [18], Huanguang (2020) [37], Tan (2024) [38], Yang (2020) [39] and Li (2024)[40]also suggest that pastoral cooperatives could become an

effective grassland management mechanism. Pastoralists within a region would merge their grasslands and livestock, collectively formulate internal management regulations [41], rotate grazing areas based on grassland quality, distribute benefits according to each household's grassland and livestock quantity, share grazing responsibilities according to household labor force and mutually supervise stocking rates and areas [38–42]. Nevertheless, this new mechanism has not been widely promoted or practiced in China. The government and other social entities have not yet participated. The actual benefits of pastoral cooperatives for pastoral areas and their impact on socio-ecological systems of pastoral regions remain to be empirically studied.

In the practice of grassland management following the implementation of the grassland contracting responsibility system, the government identified overgrazing as the primary culprit for grassland degradation [43,44]. Based on the theory that overgrazing by pastoralists leads to grassland degradation, since 2002, the government has approved and implemented grassland restoration policies such as grazing bans [45–47] and the establishment of artificial grasslands [48]. The grazing ban policy mainly targets severely degraded, unsuitable for grazing, and grasslands located in major river water conservation areas, imposing grazing prohibition and enclosure measures. However, this policy did not specify the duration of the grazing ban, nor did it provide regulations for lifting the ban [49,50]. To date, the effectiveness of the grazing ban policy has been continuously debated [51–53]. Since the implementation of the grazing ban policy in alpine pastoral regions, the ecological environment of the banned areas has recovered in a short period, but long-term bans have led to new forms of grassland degradation and decreased pastoralist incomes [54–56]. Tianyan (2022) [57] and Wang (2018) [58] believed the grazing ban policy lacks dynamic adjustment rules and standards, leading to rigidity in scope and duration.

China government has gradually established a legal system for the construction of artificial grasslands, making the development of artificial grasslands a new approach for pastoral areas [59–61]. Artificial grasslands include areas for growing forage crops for livestock and areas where natural grasslands are improved by reseeding with native and selected commercial species. Their creation aims to boost forage production, control soil erosion in degraded grasslands, and improve the ecological balance of the pasture ecosystem [62,63]. In the short term, artificial grasslands have not only become necessary for domestic ecological construction and protection but also an effective measure to address the shortage of forage in pastoral areas and the difficulty in increasing pastoralist incomes [64–66]. Yet, the construction of artificial grasslands in China started relatively late and has been in use for a short period, so their actual significance in mitigating grassland degradation and ensuring pastoralist incomes still needs to be verified.

Building on a series of issues identified within the current grassland management policy framework, this study takes China's alpine pastoral regions as its primary focus. Specifically, it examines the effectiveness of herders' cooperative organizations, whether timely adjustments to grazing prohibition policies are warranted, and the efficacy of artificial grassland interventions. By addressing these questions, we aim to develop an optimized pathway that harmonizes grassland ecological governance with the economic advancement of alpine pastoral animal husbandry, ultimately providing guidance for policy refinements and sustainable production management in these areas.

## 2. Grass-livestock social-ecological system

The grass-livestock system is a comprehensive and complex system that integrates livestock nutritional requirements and feeding behaviour, vegetation ecology, economic returns and other aspects [67,68]. For a long time, relevant studies have focused on independent research

into aspects such as vegetation growth, grassland quality, livestock performance, grazing management and the economic benefits of pastures [69,70]. However, these aspects are far from being able to study the occurrence and development of the socio-ecology of the grass-livestock system from a systematic perspective [71–73]. Given the complexity of the system, it is necessary to evaluate the grass-livestock system by integrating the knowledge of multiple disciplines such as ecology, economics and biology [74,75]. However, most of the evaluation methods are non-dynamic, fail to reflect the internal relationships, and are unable to mirror the future changing trends [76]. Faced with the dual contradictions of the degradation of grassland resources and the social and economic development in pastoral areas, stakeholders such as the government and herder households need to evaluate the management strategies and expected results.The design and implementation of effective environmental policies need to be informed by a holistic understanding of the system processes (biophysical, social and economic), their complex interactions, and how they respond to various changes [70,77].This poses certain challenges and difficulties for the establishment of predictive and evaluative models [70].

## 2.1.  Research on decision support for grass-livestock systems

There are limited research literature on carrying out decision optimization by using comprehensive evaluation methods in the fields of grass-livestock systems or pasture eco-economics. Some researchers have effectively integrated the biophysical components such as climate, soil, forage, feed, and livestock and simulated their interactions by establishing decision support models. GRAZPLAN, GrassGro and LINCFARM software have unique advantages in simulating animal production and farm management [78–80]. APSIM, EcoMod and FASSET models mainly focus on simulating the dynamics of soil nutrients, crop or forage production and the spatial variation of soil properties [81–83]. The Hurley Pasture Model is good at simulating nutrient cycling under the condition of climate change [84]. GPFARM framework can be used to predict forage production of different plant functional groups [85]. SEE and GRASIM models can determine the impacts of grazing management, such as the number of different rotational grazing plots and grazing rates, on the ecological environment and profitability [86,87]. APEX model is capable of simulating the relative differences in forage yield among different grazing treatments (grazing time, rotational grazing, grazing rates) and different soil types [88]. In these pasture simulations, the representation of herders' decision-making processes is relatively simple. There is limited room for adjusting local policies and herders' decisions in the modeling, and their application scope is not extensive. For special regions such as the alpine pastoral areas in China, the biophysical parameters and management measures of the models are not highly targeted.

Some Chinese scholars have developed a series of models applicable to Chinese family ranches by making localized adjustments to the parameters of simulation software, so as to optimize and improve the current situation of herders' production and operation as well as the grass-livestock balance [89–92]. Although the local models are highly targeted and practical, there are also problems such as a large number of parameters and difficulties in quantification. Moreover, each model exists relatively independently, resulting in issues like the lack of correlation between the data and results of different models [85]. In addition, since each component module is determined by algorithm rules and the algorithm language is rather complex, there are challenges for policy makers and stakeholders in terms of understanding, recognizing and participating in exchanges regarding the models [93].

## 2.2.  Application of system dynamics (SD) modeling in decision support for grass-livestock systems

System Dynamics (SD) modeling is one of the common methods in Integrated Assessment Modeling (IAM) [94]. It is a simulation and prediction method that can integrate various key

elements and form internal feedback [95]. The SD modeling method can provide a favorable tool for the complex grass-livestock (or pasture) systems and the decision-making needs of stakeholders [70]. It is capable of integrating multidisciplinary knowledge to explain, predict and evaluate the socioeconomic and ecological environment of the grass-livestock system [88–90]. Webb (2013) [96] combined biophysical elements such as climate, soil, vegetation growth and livestock heat stress with the operating income of pastures to develop a system dynamics simulation model and test the effectiveness of pasture adaptive schemes (the schemes include adjusting the stocking rate, improving land conditions, changing livestock breeds). Parsons (2011) [97]develop an integrated crop-livestock model to assess biophysical and economic consequences of farming practices exhibited in sheep systems.

Teague (2015) [87] integrated vegetation dynamics, livestock feeding behaviors, live weight dynamics, profits and crop feed to construct a system dynamics simulation model. Under the conditions of low to medium feeding levels and fixed or variable stocking rates, it simulated the possible ecological and economic impacts that the grazing period and the number of paddock rotations might have in the future. Ibanez (2018) [98] integrated herbage mass, forage cost, livestock prices, livestock inventory numbers and ranchers' behavioral trajectories to construct a system dynamics model, which was used to predict the future grassland quality of ranches and the income trends of the rancher group. Oniki (2018) [99]constructed a system dynamics model integrating vegetation growth, grazing rates, and income to analyze the impact of cooperative management, livestock taxation, and population growth on the socio-ecological aspects of pastures. Ibáñez (2020) [100] constructed a system dynamics model that included factors such as ranch and herd characteristics, forage production, soil erosion and markets, and studied the role of commercial grasslands in socio-economic and ecological aspects through sensitivity analysis. Martínez-Valderrama (2021) [101] integrated the climate environment and market economy subsystems to construct a system dynamics model in order to analyze the effectiveness of management measures such as herd size and supplementary feed in dealing with future arid climates. Qian (2022) [48] developed a system dynamics simulation model that includes elements such as income and grassland productivity to estimate the percentage of the area of natural grasslands and artificial grasslands required to achieve the target income. Han (2022) [76]established a system dynamics model that includes modules for forage planting and dairy cow breeding to evaluate the effectiveness of management measures such as adjusting the grain structure of the farm and forage planting on the grass-livestock balance of the farm. Although these researchers have chosen different focuses and developed various system dynamics models that integrate the socio-ecological elements of the grassland-livestock system, the driving effect and feedback of grassland public policies on the grassland-livestock system are inadequately reflected. In particular, the dynamic simulation evaluation and optimization research on how the grassland management policies in alpine pastoral areas promote the coordinated development of local socio-ecology is still in a blank state.

The reasons for using System Dynamics (SD) in the research on socio-ecological decision support for the grass-livestock system in alpine pastoral areas can be summarized as follows: (1) The SD model is a useful learning tool. Its simple and clear feedback diagrams and model language are helpful for improving the understanding of the grass-livestock system among policy makers and production decision makers. Effective policy analysis can be achieved through simulation and comparison [90]. (2) The SD model relies on the modeler's solid understanding of the actual situation on the ground [89,91]. It can simulate the local socioeconomic and livestock production situations more realistically, and the design of vegetation and livestock parameters, policies, and herders' decisions is also more targeted [102].

For the above reasons, this study chooses to use the system dynamics method to establish a socio-ecological simulation model for the grass-livestock system in alpine pastoral areas, construct the feedback interactions between the biophysical and socio-economic systems, enrich the research on system dynamics simulation in the pasture field, introduce variables related to local grassland management policies, aiming to predict and evaluate the implementation effects of policies and herders' decisions, so as to support the optimized adjustment of local policy measures.

## 3. Data sources

### 3.1. Case study context

Qilian County (Fig 1) is located in the central hinterland of the Qilian Mountains, in the northeastern part of Qinghai Province, China (37°25′15″N–39°05′18″N, 98°05′35″E–101°02′06″E), with an average elevation of 3169 meters. The average annual temperature is −1.8°C, and the annual precipitation ranges between 272.7 and 650.7 mm. The region is characterized by high altitude, cold climate, long freezing periods, and grassland with high nutritional forage content, abundant variety, and good palatability, which provides

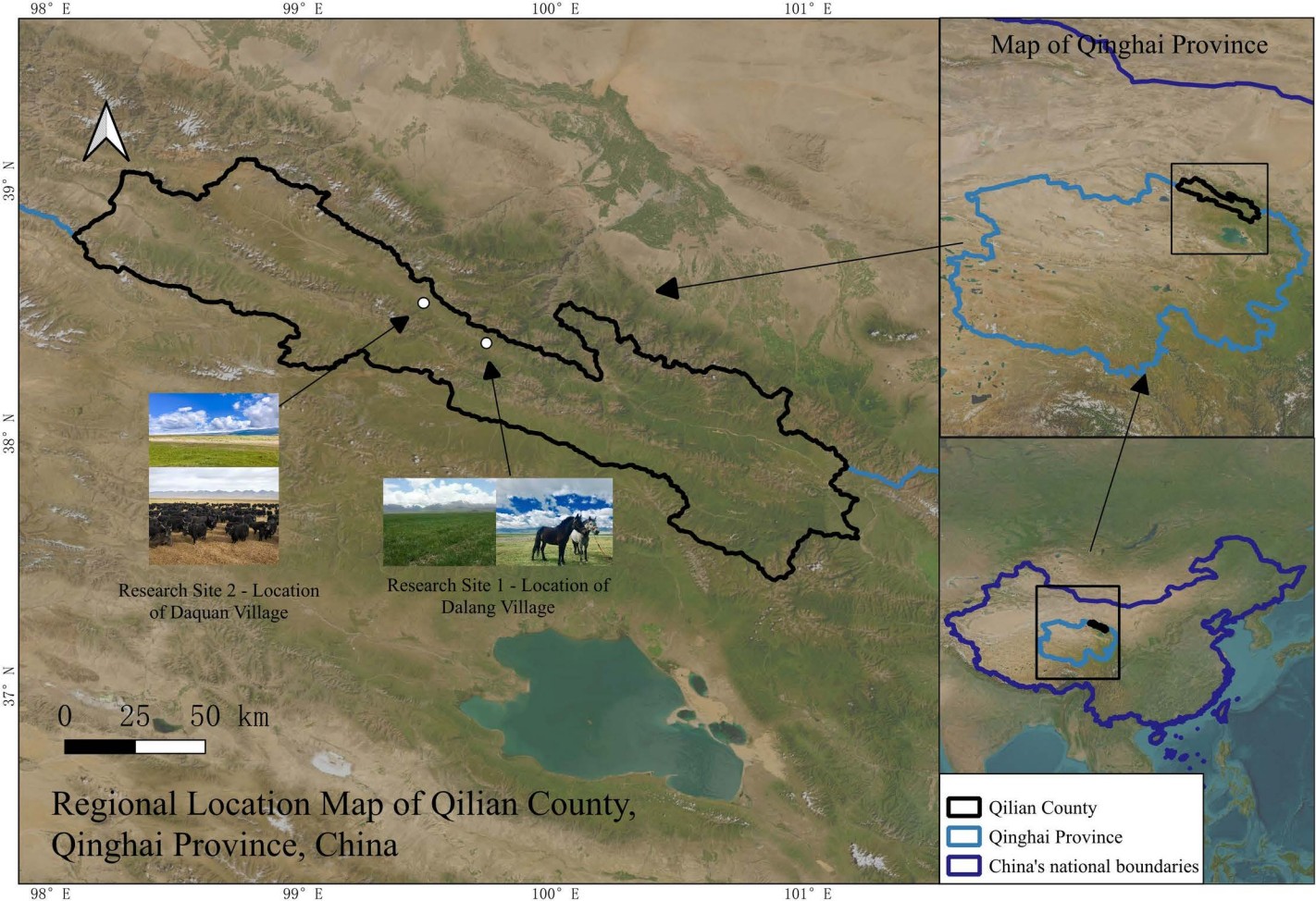

**Fig 1. The map of Qilian County, Qinghai Province, China.**

favorable conditions for the development of animal husbandry. However, the growth season for forage is short, leading to a significant forage gap in winter and spring and noticeable seasonal fluctuations in livestock weight. This area is one of China's important production bases for pastoral animal husbandry and a typical microcosm of China's alpine pastoral areas.

## 3.2. Methodology and data sources

The typical farm method is a method used to collect farm data, construct typical farms, and validate typical farm data, a concept that many researchers have applied in agricultural research [103,104]. The resource conditions, technical constraints, and policy represent the typical characteristics of a region's typical farms. This study provides a framework for analyzing the impact of public policy on households in the study area by constructing typical pastures in alpine pastoral areas. It should also be noted that this study has built a typical pasture, which blurred the identity information and personality characteristics of the pastoral households. Production and operation data adopt the average number of construction methods. The performance of livestock performance and grassland data present the common characteristics of the research area, the source is a channel for publicly available acquisition. The common decision-making characteristics formed by the herd households in the management of grazing can be summarized with the help of observation and can be verified by conducting interviews with these objective, common and observable behaviors with the pastoral households. Below is an explanation of the specific process and sample variation.

### 3.2.1. Data source and sample selection.

(1) Field research. In 2022, the author and the team conducted secondary data collection and identified public customs or notions of non-specific individuals through observation in Dalang Village and Daquan Village pastures of Yeniu Township, Qilian County. Due to multiple factors such as harsh climate conditions in the alpine pastoral areas, dispersed residences of pastoral households, and language communication barriers, the cost of transportation and time for field surveys is high, making it practically difficult to conduct large-scale comprehensive and in-depth observation. Therefore, we opted for a relatively high-quality and in-depth sample survey within feasible limits. The distribution range of pasture area in the study area is (33.33 ha ~ 285.33 ha) (mean 184.13, standard deviation 33.73, coefficient of variation 0.183), and we selected pastoral households with an area close to the mean as the focus of our investigation. First, we selected pastoral households with an area range of (150.4 ha ~ 217.87 ha) (accounting for 68% of the total pastoral households in the study area). Subsequently, we randomly selected 72 herder households for a questionnaire survey. The survey primarily focused on public data that can be publicly observed but had not been timely updated or recorded in local chronicles or official data reports. Before the survey, we explained the research purpose to the pastoral households and sought their willingness to participate to ensure informed consent and voluntariness. The survey began in September 2022. It captured the herders' operating conditions throughout the year and compared statistical data such as operating profits and livestock size from 10 years ago with the herders' actual operating status to provide data for simulation initial values and historical tests.

The questionnaire data covers: (i) Quantitative data: the area of cold and warm season pastures, the proportion of grazing-prohibited pastures, livestock scale and types, herd age structure, livestock production performance (such as weight, lambing rate, milk production, etc.), livestock production costs and benefits, household labor force, and fixed assets. Such data is typically found in secondary sources like local chronicles or provided by

herders. It is publicly observable, does not involve sensitive personal information or trade secrets, and consists of anonymous data that poses no risk of harm to participants. (ii) Qualitative data: this part of the data was obtained by the author through observing local customs without interfering with public behavior. It typically includes widely recognized information, customs, practices or popular notions that are readily observable, such as herders' traditional methods for coping with grassland degradation.

(2) Construction of Production and Management Data. To ensure data quality, we excluded households with insufficient data robustness from the 72 questionnaires, such as those newly engaged in grazing or households with reduced production capacity due to illness or disability. Ultimately, we selected 30 representative typical households that have been grazing for more than 10 years. The production and management data of typical households in the study area were constructed based on the average values.

(3) Livestock Production Performance Data. The data were sourced from questionnaires and related scientific literature, including livestock weight, lambing rates, milk yield [105–108]. By analyzing the production performance of different livestock, these data were parameterized for model construction.

(4) Grassland Biomass Data. Grassland biomass data were extracted from the long-term monitoring database of the Grass Monitoring Bureau of Qilian County, which was obtained annually through systematic sampling. This ensures the reliability and representativeness of data for different grassland types (cold season grassland, warm season grassland, and artificial grassland).

(5) Selection of Study Area. The study area includes the pastures of Dalang Village and Daquan Village, covering 602 households and 1.663 million mu (110,867.52 hectares) of grassland, divided by utilization type into cold season grassland (50%), warm season grassland (41%), and artificial grassland (8%). The response of households in this area to policy implementation and the regional characteristics are representative within the study scope.

**3.2.2. Analysis of sample variability and representativeness.** We conducted a statistical analysis of key variables (such as livestock scale, herder income) for 72 samples. The results showed that the variability of operational scale and cost-benefit data for herders in the study area is small, indicating that there is little difference in production and operational levels among herders in the study area, given the relatively small differences in the grassland area. For example, the number of livestock ranges between (500 ~ 650), with a standard deviation of 41 and a coefficient of variation of 0.067. The operational income ranges between (3 ~ 15), with a standard deviation of 0.4 and a coefficient of variation of 0.08. From the perspective of sample representativeness, constructing a typical ranch is to represent the common characteristics of a certain type of ranch, and collecting samples with small variability makes it easier to summarize general rules and patterns.

**3.2.3. Applicability of the average typical farm method.** Although the resources and operational conditions of herders exhibit variability, the average typical farm method has the following advantages: (i) Comprehensiveness and Representativeness: By selecting 30 typical herders and conducting in-depth longitudinal observation, we captured the general impact and main trends of policy implementation within the study area. (ii) Model Simplification and Operability: Facing complex variable settings, the average typical herder data provides a simplified framework, making policy impact assessments more practical.

**3.2.4. Data validation.** To ensure the reliability and authenticity of the data, we compared the survey data with Statistical Yearbook of Qilian County, Animal Husbandry Zoning of Qilian County and cross-validated it with local research literature [109] and government policy documents. Additionally, by interviewing local village officials and government management personnel, we further verified the rationality of the sample and regional selection.

# 4. Construction of grass-livestock system dynamic simulation model

## 4.1. Model description

(1) Model Assumptions. Reasonable assumptions can simplify the model and highlight the main research issues. The model assumes that all herders have homogeneous characteristics and constructs representative typical herder production and operation data. All herders have the same transhumance grazing time, meaning most local herders transfer pastures three times a year: grazing in natural cold-season pastures from January to June of the following year, grazing in natural warm-season pastures from July to September, and grazing in artificial pastures from October to December. Tibetan sheep represent local livestock, and the main local grazing livestock are Tibetan sheep and yaks. Tibetan sheep are used as the livestock unit, with one yak equivalent to four Tibetan sheep based on consumption measurement.

(2) Time Parameters. The model's simulation time step is one month. Based on the local grass growing season, which is from April to mid-August, the model simulation begins in April and ends in March of the following year. In 1993, the People's Government of Qinghai Province promulgated and implemented the 'Qinghai Province Grassland Contracting Measures', stipulating that the contracting period for grassland should not be less than 50 years. As of now, the contracting period has lasted 30 years, so it is appropriate to set the simulation period to 20 years, with the simulation starting time as April 2022 (see Table 1 for some initial values of the model).

(3) Logical relationship of each module. The model consists of three major modules: forage biomass production-livestock weight module, livestock production management module and herdsmen's business income module, with the module framework shown in Fig 2.

The model assumes that variables such as total pasture area, capital interest rate, labor wages, livestock product prices, forage prices, natural population growth rate, and climate remain constant in their initial states. The primary driving variables of the model are the production factor inputs, which affect the expected livestock numbers of herders. Consequently,

**Table 1. Initial values of the model.**

| Model Variables | Initial value | Unit |
|---|---|---|
| Weight of Tibetan sheep 25–72 months | 42 | Kg/Pc |
| Number of livestock in a typical pastoral household | 616 | One |
| Number of pastoral households in the study area | 602 | Household |
| Fixed capital of a typical pastoral household | 131125 | Renminbi Yuan (RMB) |
| Labor force of a typical pastoral household | 2.5 | One |
| Production costs of a pastoral household within one year | 0 | Renminbi Yuan (RMB) |
| Total income of a pastoral household within one year | 0 | Renminbi Yuan (RMB) |

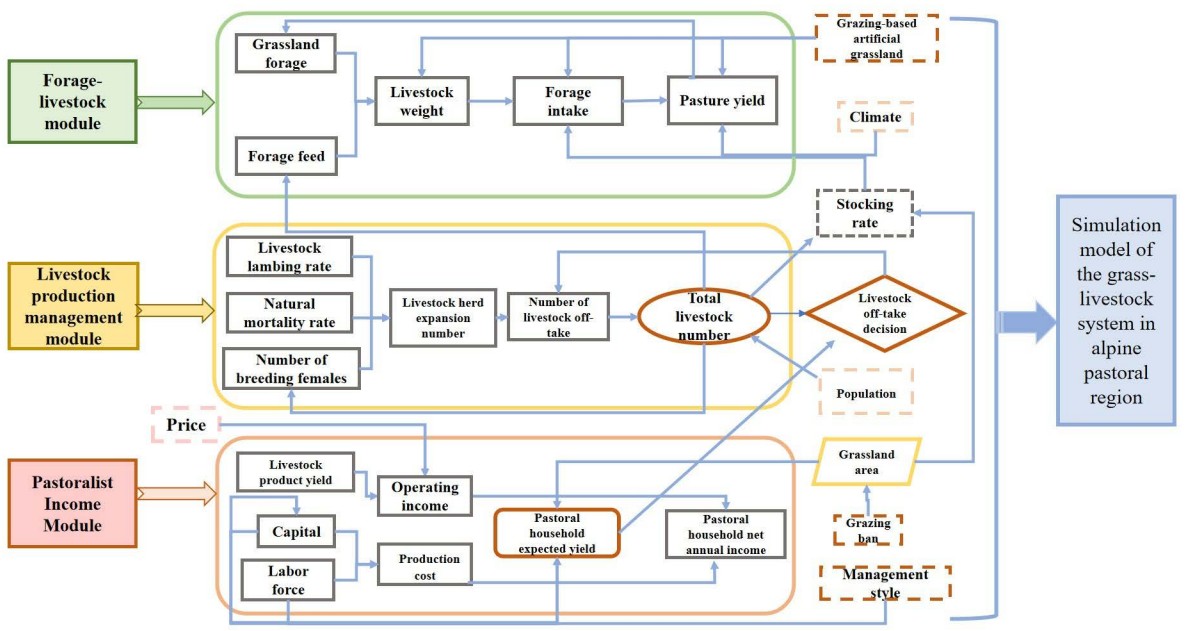

**Fig 2. Simulation model of the grass-livestock system in alpine pastoral region.**

this leads to decisions on the number of livestock to be slaughtered and changes in the livestock population size. This, in turn, triggers a series of linked changes in variables such as pasture grass production, body weight, birth rate, meat production, and total income. The model selects key variables such as grass yield in cold season pastures, livestock weight per standard livestock unit, number of standard livestock units, and average household operating profit to reflect the overall ecological and economic situation of the local grass-livestock system.

## 4.2. Model development

The model construction process fully draws upon the research findings of Parsons (2011) [93], Teague (2015) [84], Oniki (2018) [96], and Qian (2022) [98]. Among these, Parsons (2011) and Oniki (2018) provided rigorous approaches at the level of variable logic architecture, playing an extremely critical role in the construction of the variable logic system in this model. The grassland pasture yield-livestock weight module simulates the relationship between pasture yield per hectare of grassland and livestock weight; their mutual influence is primarily linked through grazing rate, intake, and degradation impact index. The livestock production management module calculates the total cost and profit of livestock production for households annually and decides productive investment based on profit. By constructing a Cobb-Douglas production function, it calculates the maximum number of grazing livestock based on labor, capital levels, and grassland area for the year, which can also be considered as the household's expected number of livestock. The livestock breeding and off-take decision module calculates the number of livestock each year; herd growth follows livestock breeding rules, is influenced by the household's expected output, and is determined by household off-take decisions. The household business income module computes the income derived from operating grasslands and livestock. The local household business income sources consist of beef and mutton sales revenue, wool sales revenue, and milk sales revenue, with beef and

mutton sales being the primary income source. The model is constructed and realized using STELLA software. A simplified logical diagram of some key variables is displayed in Fig 3, the main equations covered by the model are listed in Table 2, and specific details regarding modeling can be found in S1 Appendix.

## 4.3. Simulation scenario setting

First, the actual local conditions are set as the baseline scenario (i.e., using artificial grassland for grazing, continuously implementing grazing bans, and independent herder operations) to simulate changes in the grass-livestock system in the study area from 2022 to 2042. Next, Scenario 1 excludes artificially planted grasslands, allowing for a comparison with the baseline scenario to assess the long-term benefits of artificial grassland use. Based on the continuous use of artificial grasslands, Scenarios 2, 3, and 4 are designed as follows: Scenario 2 simulates the economic and ecological development under moderate use of grazing-ban grasslands; Scenario 3 explores the effects of implementing herder cooperative organizations; Scenario 4 examines the development trajectory of the grass-livestock system under the combined approach of moderate use of grazing-ban grasslands and cooperative management by herders.

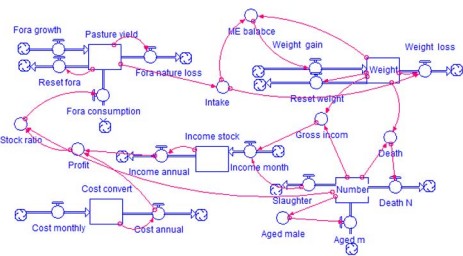

**Fig 3. Simplified diagram of the logic of the key variables of the model.**

**Table 2. Main equations of the model.**

| Equation | Number |
|---|---|
| Forage consumption = Weight×intake × Stocking rate * 30 | (1) |
| Intake = 0.00018 * pasture_yield | (2) |
| Weight gain/loss = ±0.82((Weight×Intake) × Organic_matter_digestibilitye − 0.0268 × Weight0.75)/ (0.2036 × Weight0.75) | (3) |
| Degradation effect = −1.6398 × (Cumulative_consumption/Cumulative_growth) + 1.389229 | (4) |
| Total supplement = Supplement_2 × N2f + Supplement_3 × (N3f + N1m) | (5) |
| Capital cost = Depreciation + Own capital interest | (6) |
| Cost annual = Labor Cost + Supplement Cost + Capital cost | (7) |
| Birth ratio = (Weight_3-Initial_weight)/Initial_weight × 0.404 + Birth_ratio_avg | (8) |
| Birth = (N3f + N2f) × Birth_ratio | (9) |
| Age female = Delay_2f-Decrease_3f/5 | (10) |
| Slaughter weight total = Slaughter_1m × Weight_1 + Slaughter_2m × Weight_2 + Slaughter_3m + Aged_m) × Weight_3 + (Slaughter_2f × Weight_2 + (Slaughter_3f + Aged_f) × Weight_3) × FtoM | (11) |
| Household = Initial_household_number × (1 + population_growth_rate) ^ Year | (12) |
| Adjust number of livestock = N1m × Rel_weight_1to2m + N2m × 1 + N3m × Rel_weight_3to2m + N1f × Rel_weight_1_2f + N2f × 1+ N3f × Rel_weight_3_2f | (13) |

### 4.4. Model validity test

(1) Sensitivity test of the model. The sensitivity of the model is tested by adjusting the values of key fixed parameters by ± 10% test the average range of changes in seven variables related to livestock numbers (Table 3). The results indicate that the sensitivity of all variables is less than 10%, demonstrating that the model is largely insensitive to changes in most parameters, thus fulfilling the requirements for modeling.

(2) Historical Validation of the Model. The model undergoes historical testing for the period from 2011 to 2022, comparing the simulated values for 2022 with the actual data (Table 4). The results show that the relative errors for the main variables are all less than 15%, indicating that the constructed dynamic simulation model are basically in line with the reality.

## 5. Simulation results of dynamic optimization of the grass-livestock system

### 5.1. Development trends of the grass-livestock system under the baseline scenario

The specific data variations of the simulation results can be found in S2 Appendix, and the findings are presented as follows:

(1) The average number of livestock per household increases or decreases slightly between 600.18 and 642.27 in 2022–2026, due to the low capital investment in this period, corresponding to the low expected number of livestock (Fig 4a). The average number of livestock per household is high, so there is no significant increase in the average number of livestock per household. From 2027 to 2032, the average number of livestock per household increased significantly from 621.83 to 882.65, which is due to increased productive capital investment driven by continuous profit growth over the previous five years. Consequently, this leads to an increase in the expected number of livestock by the herders and the decision to reduce the slaughter volume, thereby increasing the stock volume. After 2032, as capital investment continues to increase, the number of livestock per household remains high and shows a stable and slow growth, reaching 977.17 by the year 2042.

**Table 3. Sensitivity test of key variables.**

| Variable | Increase 10% Sensitivity mean/% | Decrease 10% sensitivity mean/ % |
|---|---|---|
| Intake ratio of Tibetan sheep | 0.55% | 0.73% |
| Birth ratio of Tibetan sheep | 7.05% | 7.27% |
| Death ratio of Tibetan sheep | 0.24% | 0.24% |
| Population growth rate | 0.26% | 0.00% |
| Invest willing ratio | 0.96% | 1.06% |
| Depreciation rate of fixed assets | 0.25% | 0.31% |
| Interest rate | 0.00% | 0.00% |

**Table 4. Model simulated value and real value error.**

| Variable | Real value | Simulated value | Error |
|---|---|---|---|
| Average Number of Livestock per Household (units) | 616.0 | 616.8 | 0.1% |
| Average Operating Profit per Household (ten thousand RMB) | 2.3 | 2.5 | 8.7% |
| Grassland Biomass | 1228 | 1080 | −12.1% |
| Livestock Weight | 42 | 48 | 14.3% |

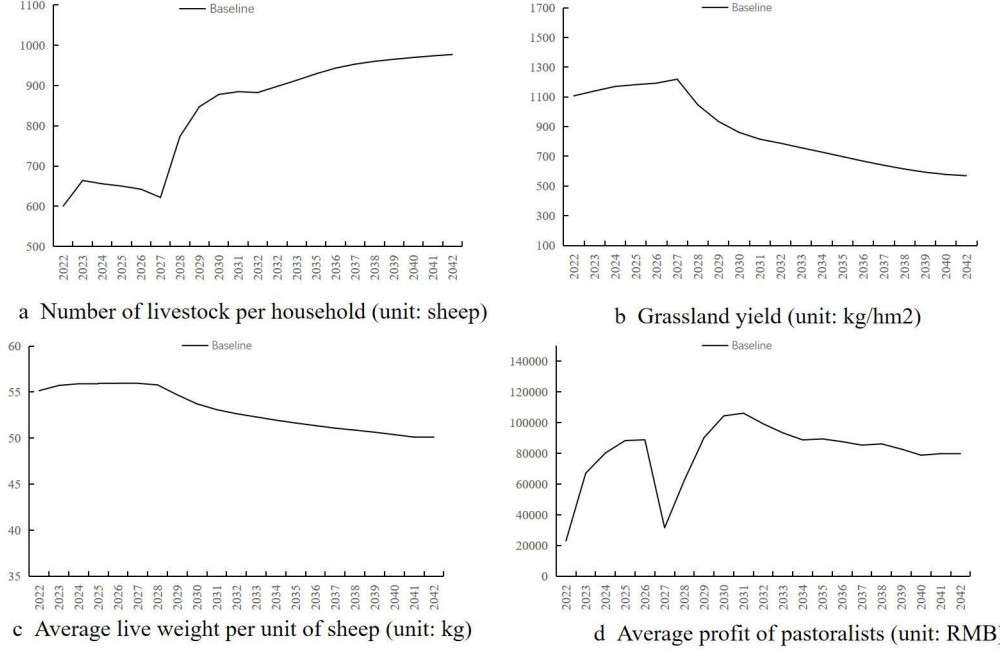

**Fig 4. Baseline scenario.**

(2) The grass yield in pastures shows an inverse trend to changes in livestock numbers (Fig 4b). From 2022 to 2027, the grass production of cool-season pastures was high and slightly increased, ranging from 1106.8 to 1219.07 kg/hm², indicating that the grazing intensity is sustainable and livestock stocking promotes the regeneration of grass. After 2027, the grass production of the cool-season pasture continued to decrease with the increase of livestock, and by 2042, it had been degraded to 568.84 kg/hm². Over the 20-year period, the grass yield during the warm season remains at a low productivity level of 400 kg/hm², with overgrazing by livestock on natural grasslands in the warm season.

(3) The average live weight of livestock shows a slow declining trend over 20 years (Fig 4c). The continuous increase in the number of livestock intensifies the grazing pressure on grasslands, leading to a yearly decline in grassland productivity. This results in insufficient grass intake for livestock, causing a decrease in live weight.

(4) The average operational profits per household rise to to 88,900 by 2026. However, due to fewer animals being slaughtered in 2027, profits drop to a low of $31,500. Between 2027 and 2031, profits increase significantly, peaking at 106,200 in 2031. Over the following decade, as livestock numbers grow modestly and their average weight declines, profits gradually decrease (Fig 4d).

## 5.2. Development trend of the grass-livestock system without establishing and utilizing artificial grasslands (scenario 1)

In the past decade, China governments have invested heavily in funds and technology to transform degraded grasslands into high-yield artificial grasslands, primarily used for grazing between October and December each year. The artificial grasslands in the study area were

established in 2017. In the short term, the establishment of artificial grasslands effectively reduced pastoralists' feed costs and increased the fertility rate of breeding livestock. However, due to the short duration of their use, there is no consensus on whether they can mitigate the degradation of cold-season natural grasslands and ensure pastoralist income, requiring several years of continued monitoring. The high initial investment necessitates evaluating the comprehensive and long-term effects of establishing grazing-type artificial grasslands in the region. To explore whether it is necessary to establish and maintain artificial grasslands locally, we simulated the future trend of the grass-livestock system without artificial grasslands. This scenario adjusts variables such as the stocking rate of cold-season natural grasslands (Stocking rate c), grazing duration, cumulative forage consumption (Forage consumption c) and intake ratio (Intake ratio a).

Simulation results (Fig 5) indicate:

(1) The average number of livestock per household increases slightly, peaking at 827.6.

(2) The pasture yield of natural grasslands fluctuates stably at a medium level of 800 kg/hm².

(3) Livestock weight decreases slightly from 55.21 kg to 49.95 kg between 2022–2026; then rapidly drops to 36.62 kg from 2027–2032, remaining low between 36–38 kg over the next decade.

(4) Pastoralist production profits remain low, dropping to 0.07 million by 2042.

The absence of artificial grasslands results in different fluctuations over time. From 2022–2026, livestock numbers change little, but feed shortages are significant, leading to a slight drop in livestock weight and low profits without significant growth. From 2025, expanding livestock numbers will exacerbate feed shortages, reducing profits as large quantities of feed

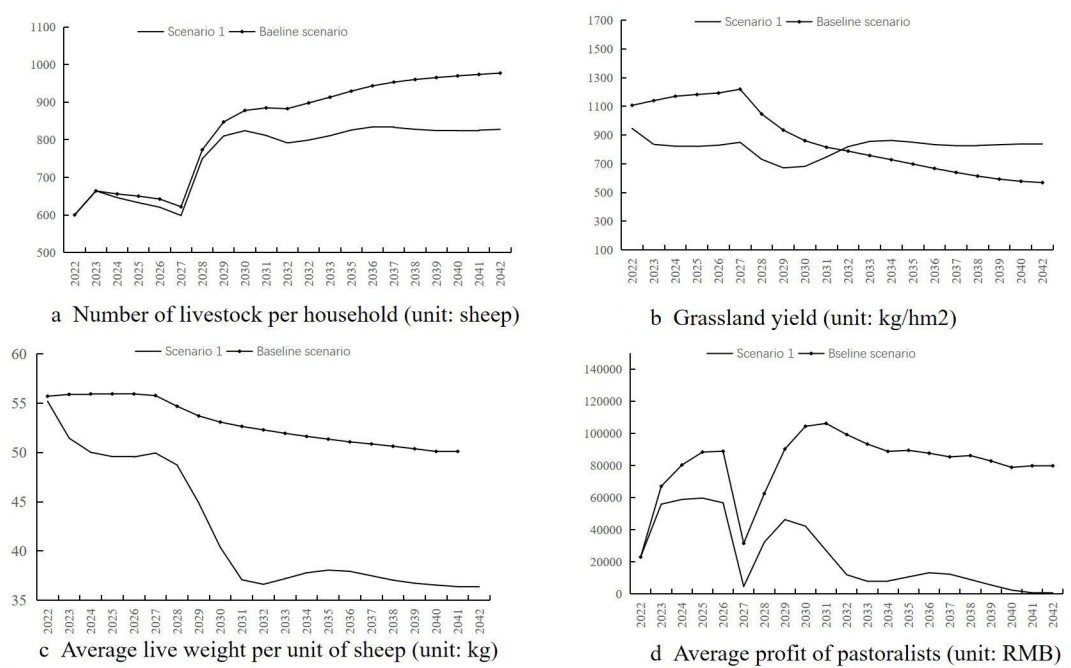

a  Number of livestock per household (unit: sheep)

b  Grassland yield (unit: kg/hm2)

c  Average live weight per unit of sheep (unit: kg)

d  Average profit of pastoralists (unit: RMB)

**Fig 5. Simulation results for scenario 1.**

are purchased, significantly decreasing livestock weight and profits again. Due to sustainable livestock numbers and low weight reducing forage consumption, the productivity of natural grasslands does not decline significantly with increased livestock numbers.

Comparing Scenario 1 with the baseline scenario shows that artificial grasslands generate multiple positive feedback for the grass-livestock system: reducing the rate of livestock weight decline and ensuring income, reducing feed purchase volume and increasing operating profits and mitigating high-intensity use of natural grasslands and their degradation.

## 5.3. Development trend of the grass-livestock system with sustainable use of grazing ban grasslands (scenario 2)

To promote the restoration of grassland ecosystems, national and local governments have implemented a series of laws and regulations related to grassland grazing bans. Since 2011, Qilian County has implemented a grazing ban on 5.11 million mu of summer pastures. To explore the impact of the grazing ban, we simulated the scenario of sustainable use of grazing ban grasslands to predict the future trend of the local grass-livestock system. This scenario adjusts variables such as the grazing prohibition policy (Grazing prohibition), cold-season grassland area (Area cold), and warm-season grassland area (Area warm) by increasing them by half of the summer grazing ban area and increasing the average grazing area per household (Area per HH).

Simulation results (Fig 6) show:

(1) The average number of livestock per household is higher than the baseline scenario, growing to 1009.8.

(2) The net primary productivity of grasslands is higher than the baseline scenario, with the cold-season natural grassland's net primary productivity decreasing to 840.58 kg/hm² after 2027.

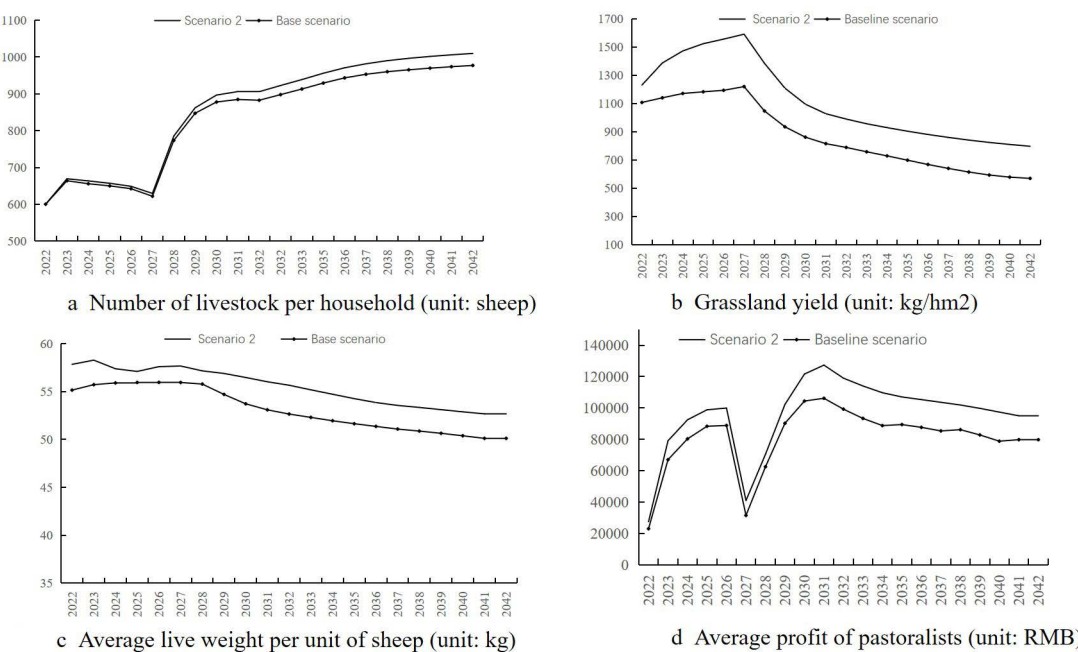

a Number of livestock per household (unit: sheep)

b Grassland yield (unit: kg/hm2)

c Average live weight per unit of sheep (unit: kg)

d Average profit of pastoralists (unit: RMB)

**Fig 6. Simulation results for scenario 2.**

(3) Livestock weight is higher than the baseline scenario, with a relatively small decline from 57.84 kg in 2022 to 52.67 kg in 2042.

(4) The average operating profit per household is higher than the baseline scenario, with an operating profit of 95,000 RMB in 2042.

The sustainable use of grazing ban grasslands has both positive and negative effects on the grass-livestock system. It increases land input elements, thus increasing the average livestock scale per household and putting pressure on natural grasslands. However, opening grazing bans expands the grazing area, alleviating pressure on natural grasslands, reducing feed purchases, and improving pasture yield, livestock weight, income and profit.

Compared to the baseline scenario, the results show that lifting the grazing ban increases land input factors. Although it slightly increases the average livestock per household, it also increases the available forage for livestock. Overall, it reduces the grazing pressure on natural grasslands, decreases feed costs, increases livestock weight, and improves profits. The positive effects of lifting the grazing ban outweigh the negative effects. Therefore, the current grazing ban policy may not be entirely effective and might even produce negative effects. If grazing ban grasslands are scientifically allocated and rotational grazing periods are reasonably set, moderate use of grazing ban grasslands can help balance the ecological and production functions of the grass-livestock system and improve the relationship between ecological protection and pastoralist interests.

## 5.4. Development trend of the grass-livestock system under the implementation of pastoral cooperative organizations (scenario 3)

Since 1993, Qinghai Province has implemented the grassland contracting responsibility system, and the study area now mainly consists of scattered individual pastoralists as production units. With the proposal of the concept of "collective grassland governance and joint livestock operation," "pastoral cooperative operation" has been considered an ideal supplementary governance mechanism to the grassland contracting responsibility system.

Based on the author's data obtained through observation without interfering with public behavior, local village committee leaders cooperative directors, and village representatives have been promoting the establishment of 'pastoral cooperative organizations' in recent years. However, due to uncertainties about the economic impact of this new mechanism, there has been no consensus on joint operation among pastoralists. Although the primary goal of the governance mechanism of pastoral cooperative organizations is to address grassland degradation and achieve ecological benefits, ensuring stable economic benefits is crucial for the sustainability of this mechanism.

To explore whether pastoral cooperative organizations contribute to the coordinated development of the ecological economy in pastoral areas, we simulated the production and ecological changes of the grass-livestock system under the scenario of pastoral cooperative organizations. This scenario primarily adjusts investment decision variables (Corporative pasture management). If the profit difference (d Profit) of pastoralists over the past two years is negative, a fixed ratio (Invest willing) of operating profit is reinvested into production; otherwise, no additional productive capital (Invest decision) is invested.

Simulation results (Fig 7) show:

(1) The average number of livestock per household is lower than the baseline scenario, reaching 818.55 by 2042.

(2) The net primary productivity of natural grasslands is higher than the baseline scenario, although the cold-season grassland pasture yield still decreases to 780.08 kg/hm².

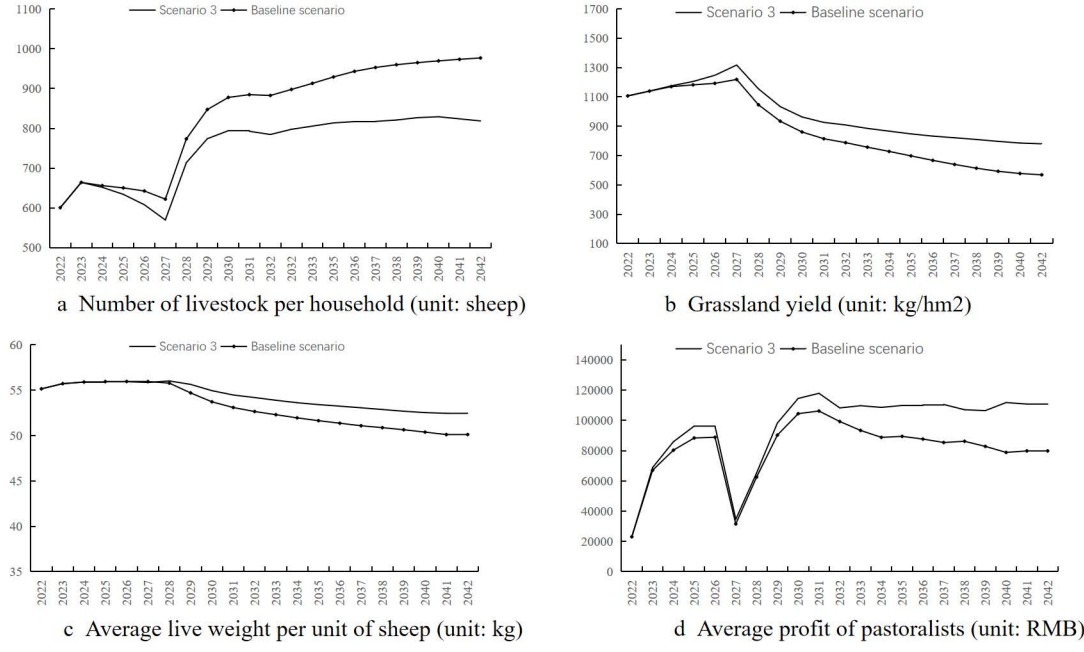

**Fig 7. Simulation results for scenario 3.**

(3) Livestock weight shows a slight decline, reaching 52.45 kg by 2042, but remains higher than the baseline scenario.

(4) The average operating profit per household exceeds 108,500 yuan from 2030 onwards, significantly higher than the baseline scenario.

Under the scenario of pastoral cooperative organizations, the limitation on productive capital investment reduces livestock scale expansion, maintains better grassland and livestock conditions, lowers feed costs, and results in higher profits than the baseline scenario.

Compared with the baseline scenario, the results indicate that the advantages of implementing pastoral cooperative organizations mainly lie in reducing the average number of livestock per household and ensuring income. Thus, by pooling production factors and jointly deciding on productive investments, it is possible to reduce production scale per household, achieving both economic income and grassland ecological benefits. This study's results align with other related research, suggesting that pastoral cooperatives facilitate integrated grassland-livestock management and rotational grazing, thereby reducing grassland carrying pressure. Pastoral cooperatives can also achieve specialized, standardized, and scaled operations through integrated production resources, ensuring pastoralist income.

## 5.5. Development trend of the grass-livestock system under combined management policies (scenario 4)

The above simulation results indicate that the implementation of a single policy or management mechanism does not achieve the best ideal effects. Whether it is just establishing grazing-type artificial grasslands, moderately utilizing grazing ban grasslands, or implementing pastoral cooperative organizations, the pasture yield and profits show varying degrees of

decline and do not reach optimal levels. Therefore, we simulate the impact of combining these management policies on the grass-livestock system. This scenario involves continuing the current use of artificial grasslands, implementing the policy of lifting grazing bans, and adopting pastoral cooperative management, predicting changes over the next 20 years.

Simulation results (Fig 8) show:

(1) The average number of livestock per household increases to only 866.31, effectively controlling livestock numbers.

(2) The net primary productivity of grasslands is higher than the baseline scenario, with a decrease to 952.23 kg/hm² by 2042, but productivity remains at a medium-high level.

(3) Livestock weight remains between 55–58 kg.

(4) The average operating profit per household remains high, around 130,000 yuan from 2031 to 2042.

Compared to the above scenario, scenario 4 effectively controls the increase in the average number of livestock per household, maintains high livestock weight, and achieves the highest operating profit with a slow growth trend. Over the 20 years, the pasture conditions remain good, with cold-season pasture yield consistently at or above 952.23 kg/hm². Therefore, the combination of maintaining the existing artificial pasture area and yield, sustainable utilizing grazing ban grasslands and implementing pastoral cooperative mechanisms shows that over the next 20 years, the average number of livestock per household will increase slowly, pastoralist income is secured, pasture quality remains good and the local grass-livestock system achieves coordinated ecological and economic development.

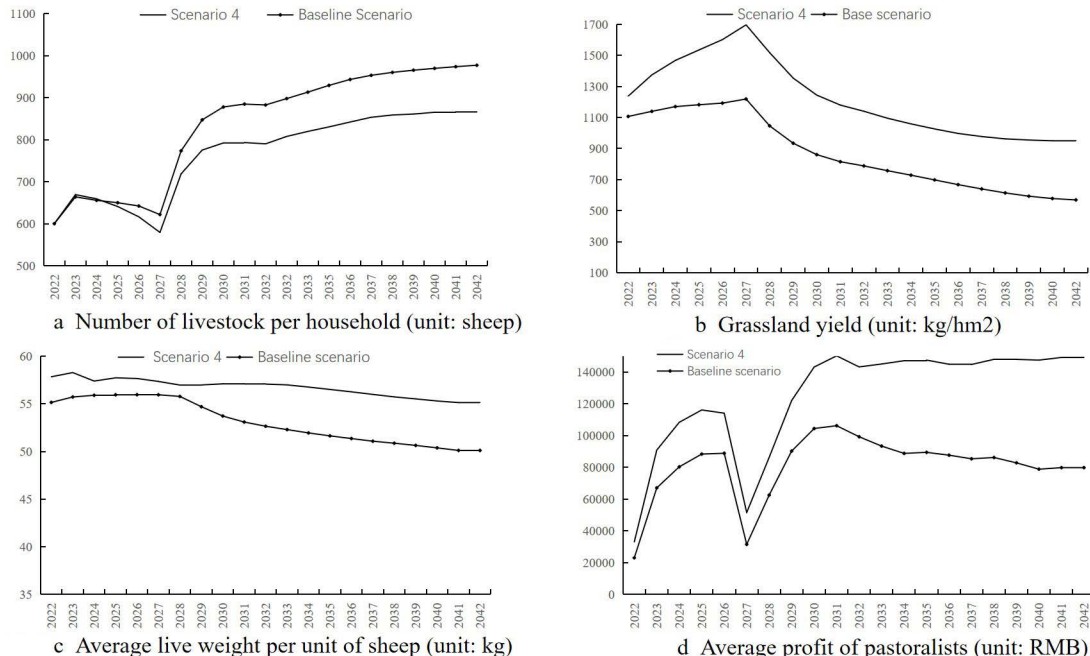

a  Number of livestock per household (unit: sheep)

b  Grassland yield (unit: kg/hm2)

c  Average live weight per unit of sheep (unit: kg)

d  Average profit of pastoralists (unit: RMB)

**Fig 8. Simulation results for scenario 4.**

## 6. Conclusion and recommendations

The study constructed a dynamic optimization simulation model of a grass-livestock system, incorporating grassland ecological and livestock physiological parameters into the economic production of pastoral households, to provide early warnings for the sustainability of production and ecology in alpine pastoral regions. Through scenario analysis, the study simulated and compared the expected effects of different policies and management practices, and further simulated the comprehensive effects of combined management policies.

The research results indicate:

(1) Artificial grasslands effectively alleviate the degradation of natural grasslands and ensure livestock production and pastoral economic stability within a certain period [110]. Therefore, local efforts must continue to strengthen the follow-up management of artificial grasslands to sustain their beneficial effects.

(2) The reasonable use of no-grazing grasslands can, to some extent, alleviate the degradation of natural grasslands, maintain livestock weight, and ensure pastoral household income[111]. The increased input of land elements from using no-grazing grasslands leads to an increase in the average production scale per household and a corresponding increase in profits. At the same time, the use of no-grazing grasslands expands the total grazing area without causing excessive tension on grassland resources, indicating that the positive effects of using no-grazing grasslands outweigh the negative effects. This suggests that if reasonable resting periods and rotational grazing systems can be established, the positive effects of lifting grazing bans outweigh the negative ones, benefiting the balance between ecological protection and pastoralist interests.

(3) Establishing and perfecting internal cooperative management regulations in pastoral areas, the growth of livestock numbers is effectively controlled by pastoral cooperative organizations, resulting in lower grassland degradation levels and more balanced annual pastoral household incomes [39]. The cooperative management of livestock, centralized use of grasslands and the formulation of internal management contracts by cooperative organizations reduce the input of labor and fixed capital, thereby reducing the average production scale per household. This demonstrates that mature pastoral cooperative organizations contribute to the sustainable development of pastoral areas.

(4) Given the premise of the sustainable use of local artificial grasslands, the combined implementation of reasonable use of no-grazing grasslands and the establishment of pastoral cooperative organizations is more effective [112,113]. Compared to the above four simulated scenarios, the simulation results of the combined policy management show that livestock numbers and weight, pastoral household income and pasture yield all remain in a sustainable state.

Based on the conclusions of the above studies, at this stage, in the process of promoting the sustainable development of animal husbandry in alpine pastoral areas, it is essential for the grassland livestock industry in these regions to advance grassland ecological management and livestock development simultaneously. This involves ecological restoration and governance alongside local livestock development, ensuring ecological protection during development. The following recommendations are specifically proposed:

(1) Reasonably assess the duration of grazing bans. Apply smart dynamic system simulations to evaluate suitable enclosure periods for alpine grasslands. Implement adaptive grazing management strategies tailored to local conditions, such as adaptive dynamic stocking rates, mixed grazing ratios, rotational grazing rules, and resting periods. Establish a

dynamic balance between biological and environmental factors to achieve ecological protection in alpine pastoral regions.

(2) Establish cooperative management mechanisms in pastoral areas. Design internal operational mechanisms and principles for pastoralist cooperative organizations according to local conditions. Develop unique and advantageous industries in alpine pastoral regions through cooperative operations, increasing the added value of livestock products to raise pastoral household incomes, thereby reducing the expansion of livestock numbers by households.

(3) Enhance investment in grassland protection and construction. Actively adopt rotational grazing and enclosure, supplementary sowing, and improvement measures. Employ a series of grassland restoration techniques, planting grass in degraded areas with suitable water and heat conditions. For established artificial grassland areas, manage artificial grasslands scientifically and meticulously to extend their usable lifespan.

Overall, this study focuses on the policy research area of the coordinated development of the ecological economy in alpine pastoral areas, using system dynamics methods to construct an IAM for the grass-livestock system. The modelers gain a deep understanding of the local actual conditions, covering aspects such as cost-benefit analysis, livestock production decision-making, seasonal variations in grassland ecology, forage consumption, and its relation to livestock performance. They fully integrate the demands and concepts of both policy implementers and pastoralists—the two key stakeholders—using qualitative data, aiming to present the current policies and proposed optimization paths to pastoralists and government staff to facilitate consensus on grassland management. However, it should be noted that the model design does not include random variables, and given the uncertainties arising from climate change and market economic fluctuations, further optimization and improvement are still needed.

## Supporting information

**S1 Appendix. Model Construction of the Dynamic Simulation for the Grass-Livestock System in China's Alpine Pastoral Areas.**
(DOCX)

**S2 Appendix. Data on the dynamic changes in average profit of pastoralists, livestock weight, numbers and grassland forage yield from 2022 to 2042.**
(XLSX)

## Author contributions

**Conceptualization:** Ligan Cai, Jian Chen.

**Data curation:** Ligan Cai, Junhao Zhao.

**Formal analysis:** Ligan Cai, Junhao Zhao, Jian Chen.

**Methodology:** Ligan Cai, Jian Chen.

**Project administration:** Jian Chen.

**Resources:** Jian Chen.

**Software:** Ligan Cai.

**Supervision:** Jian Chen.

**Writing – original draft:** Ligan Cai, Junhao Zhao, Jian Chen.

**Writing – review & editing:** Ligan Cai, Junhao Zhao, Jian Chen.

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
