## [Decision Letter · Decision Letter 0]

6 Oct 2024

PONE-D-24-30315Coevolutionary Dynamics in the Grass-Livestock Social-Ecological System of China's Alpine Pastoral Areas: A Case Study of the Qilian Mountains Region in ChinaPLOS ONE

Dear Dr. Chen,

Thank you for submitting your manuscript to PLOS ONE. After careful consideration, we feel that it has merit but does not fully meet PLOS ONE’s publication criteria as it currently stands. Therefore, we invite you to submit a revised version of the manuscript that addresses the points raised during the review process.

One or more reviewers recommended to add some citations. Please carefully review their suggestions and only add the suggested reference(s) when they are relevant to the manuscript. 

We look forward to receiving your revised manuscript.

Kind regards,

Agung Irawan

Academic Editor

PLOS ONE

Journal requirements: 1. When submitting your revision, we need you to address these additional requirements.Please ensure that your manuscript meets PLOS ONE's style requirements, including those for file naming. The PLOS ONE style templates can be found at https://journals.plos.org/plosone/s/file?id=wjVg/PLOSOne_formatting_sample_main_body.pdf and https://journals.plos.org/plosone/s/file?id=ba62/PLOSOne_formatting_sample_title_authors_affiliations.pdf 2. Please match your authorship list in your manuscript file and in the system. 3. You indicated that ethical approval was not necessary for your study. We understand that the framework for ethical oversight requirements for studies of this type may differ depending on the setting and we would appreciate some further clarification regarding your research. Could you please provide further details on why your study is exempt from the need for approval and confirmation from your institutional review board or research ethics committee (e.g., in the form of a letter or email correspondence) that ethics review was not necessary for this study? Please include a copy of the correspondence as an ""Other"" file.

Reviewers' comments:

Reviewer's Responses to Questions

**Comments to the Author**

1. Is the manuscript technically sound, and do the data support the conclusions?

Reviewer #1: Partly

Reviewer #2: Yes

2. Has the statistical analysis been performed appropriately and rigorously?

Reviewer #1: I Don't Know

Reviewer #2: Yes

3. Have the authors made all data underlying the findings in their manuscript fully available?

Reviewer #1: No

Reviewer #2: Yes

4. Is the manuscript presented in an intelligible fashion and written in standard English?

Reviewer #1: Yes

Reviewer #2: Yes

5. Review Comments to the Author

Reviewer #1: the paper is based on a bio-economic model built on an averaged typical farm performance. The objective is to simulate impacts of livestock according to different scenarios. I have three major points of concern:

1) The research is potentially interesting but lacks reference to a theoretical framework and research data is somewhat poorly supported. In the title and introduction reference to socio-ecological system is reported, but actually the work is based on a bio-economic model. The most important issue is that the authors do not include any kind of stakeholder participatory round. You may refer to this paper to understand what a socioecological approach is (one of many examples) Elsawah, S., Pierce, S.A., Hamilton, S.H., Van Delden, H., Haase, D., Elmahdi, A. and Jakeman, A.J., 2017. An overview of the system dynamics process for integrated modelling of socio-ecological systems: Lessons on good modelling practice from five case studies. Environmental Modelling & Software, 93, pp.127-145.

2) The results corroborate general statement about the need to support cooperative management and sustainable use of grassland resources but I'm wondering if we need a model to support this general statement. The authors should add more details on the data that they used, how was the questionnaire built, haw many years of grass data collection have been performed. The 'mechanics' of the model seems fine but results really depend on the quality of data.For instance, we do not have a clue about the variability of your sample. Ths, it is complicate to understand how an averaged typical farm can be considered a valid approach in the model.

3) the references to existing data, theoretical framework, etc. is not sufficient and is a clear weakness of the manuscript.

some more specific comments as following:

L50 intro an indication if ha of pastoral area would be useful as you mention significant influence at globe level

L57 reasons for degradation of these ecosystems?

L62 state is a bit too generic. Is there a specific agency or ministry? A specific law framework?

L81 the first reference appears here after a number of statements. The information reported should be supported by appropriate references.

L99 'Violating pastoralists' please, clarify the meaning

L100 'this new mechanism of joint management can solve the "tragedy of the commons": but at L75 you have already said that ' This policy, by defining property rights, effectively solved the "tragedy of the commons'. In addition, the 'tragedy of the commons' deserves a proper definition and you have not indicated references.

L120-124 here you define artificial grassland but I find the description not informative. What is it? Forage crops? renaturalization with commercial seeds? Which species are employed? Statements like 'Artificial grasslands mainly include artificial grasslands for controlling degraded grasslands and semi-artificial grasslands for improving and reseeding natural grasslands' does not help to understand this practice.

L134 Please, add a brief explanation (with references) of what a system dynamic approach is.

L130-158 Here you mix gaps in the state of the art and the objectives, but it is not very clear what are the problems and how you intend to tackle them. I would appreciate to have a clear paragraph on what are the problems, gaps of research etc. and then a paragraph to explain the objectives of the study.

L190 which % of total households do the 72 households cover?

L191 '30 typical households were selected from these 72 households' based on what?

L202 which experimental data? Reference?

table 1 point 4 'pen and circle' What is it?

L215 'ONIKI S and PARSONS' why uppercase?

L218 Yak is a small ruminant?

Fig 3 I suggest to put this in appendix and add here a more simplified figure to explain the model.

Fig 4, 5 and 6 idem

L267 'Relevant theories' which theories? Defined by whom?

L325 'research' Which research?

L373 'research' Which research?

in general, I'd reduce consistently Chapter 3. The full description can be moved in an appendix.

Reviewer #2: General Comments

This paper uses a system dynamics approach to study the co-evolutionary dynamics of the grass-livestock socio-ecological system in the Qinghai-Tibet Plateau of China. Using the Qilian Mountains as a case study, a simulation model of the grass-livestock system was constructed to analyze the long-term impacts of different management strategies (e.g., sustainable use of artificial grasslands, rational use of grazing bans, pastoral cooperative management) on grassland ecology and pastoral economy. The study found that reasonable use of fenced grasslands, combined with cooperative management, can effectively control livestock numbers, ensure pastoralists' income, and maintain grassland quality. It is recommended to promote the coordinated development of grassland ecological management and pastoral economic development by adopting grassland restoration technologies, strengthening the management of artificial grasslands, and fostering the establishment of pastoral cooperative organizations. Therefore, this study is worthy of publication but requires some revisions.

Main Comments

1. Many studies have explored the effects of government management on the Qinghai-Tibet Plateau (e.g., doi:10.1016/j.scitotenv.2024.176404; doi:10.3390/plants12183182; DOI: 10.3389/fpls.2022.991287; DOI: 10.1002/ldr.3835), but this paper did not mention the contributions of these studies. It is recommended to supplement the literature review to better position the innovation of this study.

2. There are too many figures in the main text. It is suggested to merge or delete some figures to focus on presenting the most important results.

3. The number of references is relatively small. It is recommended to add more relevant high-quality references to enhance the scientific rigor of the study.

Minor Comments

1. The figures are rather rough, especially Figure 1. It is recommended to further refine them.

2. The description "See S1" is not specific enough; it is recommended to clarify it.

3. Figure 3 only has a title; it is suggested to add a brief explanation in the main text.

4. The text in Figures 7, 8, 9, and 10 is too small, and it is recommended to enlarge the font.

5. The spelling of "Decisio" is incorrect and should be changed to "Decision."

6. The use of "livestock number" and "livestock numbers" should be consistent.

7. The sentence "the number of livestock remain high" should be changed to "the number of livestock remains high" to ensure subject-verb agreement.

6. PLOS authors have the option to publish the peer review history of their article (what does this mean? ). If published, this will include your full peer review and any attached files.

**Do you want your identity to be public for this peer review?** For information about this choice, including consent withdrawal, please see our Privacy Policy .

Reviewer #1: No

Reviewer #2: No

---

## [Author Response · Author response to Decision Letter 0]

19 Dec 2024

Response to Reviewers for manuscript PONE-D-24-30315

Dear Editors and Reviewer:

On behalf of all contributing authors, I would like to express our sincere appreciation for your letter and the reviewer’s constructive comments concerning our paper entitled “Coevolutionary Dynamics in the Grass-Livestock Social-Ecological System of China’s Alpine Pastoral Areas: A Case Study of the Qilian Mountains Region in China” (PONE-D-24-30315). These comments are valuable and helpful for improving our paper. In response to the editor and reviewers' comments, we have made extensive modifications to our manuscript to enhance the credibility of our results.

This letter is divided into three main parts. The first part addresses the journal's requirements, the second part responds to Reviewer 1’s comments, and the third part responds to Reviewer 2’s comments.

In the revised version of the manuscript, all changes to our manuscript are highlighted in yellow-shaded text. Point-by-point responses to the editor and reviewers are listed below this letter.

lf there are any other modifications we could make, we would like very much to modify them and we really appreciate your help.

Sincerely,

Ligan Cai, Junhao Zhao, Jian Chen

1. Responses to journal requirements:

Response:

Thank you very much for your reminder. We have made corrections according to the required format.

(2). Please match your authorship list in your manuscript file and in the system.

Response:

Thank you very much for your reminder. We have reviewed the author list as required.

(3). You indicated that ethical approval was not necessary for your study. We understand that the framework for ethical oversight requirements for studies of this type may differ depending on the setting and we would appreciate some further clarification regarding your research. Could you please provide further details on why your study is exempt from the need for approval and confirmation from your institutional review board or research ethics committee (e.g., in the form of a letter or email correspondence) that ethics review was not necessary for this study? Please include a copy of the correspondence as an “Other" file.

Response:

Thank you very much for your reminder and understanding. We have briefly described this in the clarification letter and provided confirmation from the Research Ethics Committee that no ethical review was required for this study. Since the first author’s affiliated institution at the time of writing this paper did not have an Institutional Review Board, a well-known psychiatric hospital in China was entrusted to conduct the review, and they issued a confirmation that ethical review was not required.

2. Response to reviewer 1’s comments:

Reviewer #1commented：(1) The research is potentially interesting but lacks reference to a theoretical framework and research data is somewhat poorly supported. In the title and introduction reference to socio-ecological system is reported, but actually the work is based on a bio-economic model. The most important issue is that the authors do not include any kind of stakeholder participatory round. You may refer to this paper to understand what a socioecological approach is (one of many examples) Elsawah, S., Pierce, S.A., Hamilton, S.H., Van Delden, H., Haase, D., Elmahdi, A. and Jakeman, A.J., 2017. An overview of the system dynamics process for integrated modelling of socio-ecological systems: Lessons on good modelling practice from five case studies. Environmental Modelling & Software, 93, pp.127-145.

Response:

We sincerely appreciate your insightful comments, which have significantly contributed to improving our manuscript. Below, we address each of your suggestions:

a.Regarding the lack of theoretical framework:

We acknowledge the insufficiency in referencing a theoretical framework. With guidance from your suggested literature, particularly Elsawah et al. (2017), we have further explored integrated assessment models (IAMs) within social-ecological systems research. We have systematically reviewed relevant literature on the grass-livestock integrated assessment model and expanded Section 2 of the paper, now titled ‘Grass-Livestock Social-Ecological System’. This section clearly outlines the concepts of grass-livestock systems, their role in decision-support studies, and the application of system dynamics (SD) modeling within such systems.

b.Regarding the research data's insufficient support:

We revisited and restructured our data sources to strengthen the empirical support for our arguments. In Section 3.2.1: Data Source and Sample Selection, we provide a detailed description and categorization of data sources. Through a survey, we collected data on grassland area types, livestock scales and species, herd age structures, and livestock performance indicators (e.g., body weight, lambing rates, milk production, and mortality). Additional data on production costs, household labor, and fixed assets were obtained from 72 questionnaires, of which 30 were used for constructing representative pastoral household profiles based on averages. Livestock performance data, including intake rates and digestibility, were sourced from relevant literature.

c. Regarding alignment between the title, introduction, and the model:

We understand your concern regarding the discrepancy between the title, introduction, and the bio-economic model used in our study. In response, we have clarified that integrating grassland ecological indicators and livestock biological performance better reflects the ecological-economic interactions within the grass-livestock system. The newly expanded Section 2 explicitly frames the grass-livestock system as a complex system incorporating nutritional needs, grazing behavior, vegetation ecology, and economic outcomes. We have further emphasized in Section 2.2 the role of system dynamics (SD) modeling in integrating ecological, biological, and economic parameters for decision-making. This ensures a stronger coherence between the title, introduction, and core content of the paper.

d. Regarding the absence of stakeholder participation:

We fully acknowledge the importance of incorporating stakeholder perspectives, as you highlighted. In fact, during the fieldwork phase, we conducted in-depth observations of key stakeholders, including herding families and government officials, in a manner that did not interfere with or influence their public behavior. These observations include their daily decision-making processes, their intentions to cooperate, and their ways of addressing grassland degradation, and are often representative of prevailing local practices and traditional customs. It could also be noted that this study has built a typical pasture, which blurred the identity information and personality characteristics of the pastoral households. Production and operation data adopt the average number of construction methods. The performance of livestock performance and grassland data present the common characteristics of the research area, the source is a channel for public available acquisition. The common decision-making characteristics formed by the herd households in the management of grazing can be summarized with the help of observation and can be verified by conducting interviews with these objective, common and observable behaviors with the pastoral households. We integrate this data into model development and refinement to better reflect real-world macro-phenomena and stakeholder priorities. Our study is also based on these anonymized data in order to construct a System Dynamics (SD) model of grassland ecology and pastoralism in alpine pastoral areas, and that SD model focuses more on specifying general scenarios and typical changes in Social-Ecological systems rather than focusing on the micro-level behavior of individual pastoralists.

Reviewer #1commented:（2）The results corroborate general statement about the need to support cooperative management and sustainable use of grassland resources but I'm wondering if we need a model to support this general statement. The authors should add more details on the data that they used, how was the questionnaire built, haw many years of grass data collection have been performed. The 'mechanics' of the model seems fine but results really depend on the quality of data.For instance, we do not have a clue about the variability of your sample. Thus, it is complicate to understand how an averaged typical farm can be considered a valid approach in the model.

Response:

Thank you for your thoughtful comments and for highlighting key areas for improvement. We greatly appreciate your constructive feedback.

a. Regarding the necessity of the model to support general statements:

While it may appear that our results confirm general assertions about sustainable grassland resource use, the model serves a critical role in providing a structured analytical framework. Rather than restating general conclusions, our model delves into the intricate relationships and dynamic interactions among various elements within the grassland resource system. By simulating the long-term effects of different management strategies, such as changes in ecological conditions, livestock productivity, and socio-economic factors, the model offers precise, detailed, and forward-looking insights. This depth and scope far exceed what general statements can achieve and significantly support policymakers in formulating targeted policies and optimizing practical solutions.

b. Regarding additional data details:

We fully agree with your suggestion and will add more specific details about our data collection in Section 3.2: Data Source and S1 Appendix. This will include details on the questionnaire design, structure, and the years of data collection.

c. Regarding sample variability:

We recognize your concerns about the representativeness of an average typical farm. In our study, the construction of a "typical pastoral household" reflects common characteristics in livestock performance, grassland biomass, and household decision-making behaviors, minimizing variability in these factors. The primary variability stems from differences in grassland area. To address this, we conducted a variability analysis focusing on pasture size. The grassland area in our study region ranges from [33.33 ha～285.33 ha] (mean 184.13, standard deviation 33.73, coefficient of variation 0.183). We selected households with pasture sizes close to the mean, specifically within the range of [150.4 ha～217.87ha] , resulting in 72 representative households. This approach reduces variability in production and operational data, enabling us to construct a more representative typical household. Further details are now clarified in Section 3.2.2: Analysis of Sample Variability and Representativeness.

Reviewer #1commented：（3）the references to existing data, theoretical framework, etc. is not sufficient and is a clear weakness of the manuscript.

Response:

Thank you for pointing out this weakness. To address this, we have conducted a comprehensive review of relevant literature and have systematically integrated these references into the manuscript.

Reviewer #1commented：（4）some more specific comments as following:

“L50 intro an indication if ha of pastoral area would be useful as you mention significant influence at globe level”

Response:

Thank you for your suggestion. We have added the following information: ‘China's 400 million hectares of pastoral land rank as the world's second-largest, covering approximately 40% of the country. These lands are located in the drier, higher regions of north and northwest China and are primarily inhabited by various ethnic minorities.’

Reviewer #1commented：L57 reasons for degradation of these ecosystems?

Response:

Thank you for your valuable suggestion. We have added the following: ‘The transformation of China’s alpine pastoral areas is currently facing many challenges, especially due to the irrational number of grazing animals and the imperfect system of grassland resource management, the alpine pastoral areas are faced with the problem of the continuous deterioration of grassland ecology.’

Reviewer #1commented：“L62 state is a bit too generic. Is there a specific agency or ministry? A specific law framework?”

Response:

Thank you for your suggestion. We have added the following: ‘The State Council, the Ministry of Agriculture and Rural Affairs, the State Forestry and Grassland Administration, and other departments have gradually established a grassland management system. This system is based on the grassland contract responsibility system, grazing bans, artificial grassland construction, and other major projects, with supporting laws, regulations, and policy measures implemented.’

Reviewer #1commented：“L81 the first reference appears here after a number of statements. The information reported should be supported by appropriate references.”

Response:

Thank you for your comment. Relevant references have now been cited.

Reviewer #1commented：“L99 'Violating pastoralists' please, clarify the meaning”

Response:

Thank you for your insightful comment. The original sentence, "Violating pastoralists" means pastoralists who violate internal management rules, has been removed in the revised version as we believe it was redundant and did not add value to the context.

Reviewer #1commented：“L100 'this new mechanism of joint management can solve the "tragedy of the commons": but at L75 you have already said that ' This policy, by defining property rights, effectively solved the "tragedy of the commons'. In addition, the 'tragedy of the commons' deserves a proper definition and you have not indicated references.”

Response:

Thank you for your thoughtful suggestion. The reference to the "tragedy of the commons" has been removed in the revised version, as this phenomenon has not been observed in alpine pastoral areas. The grassland contract system was primarily introduced to enhance household production and facilitate management. We believe that mentioning the "tragedy of the commons" is not essential in this context and does not impact the readers' understanding of the research background.

Reviewer #1commented：“L120-124 here you define artificial grassland but I find the description not informative. What is it? Forage crops? renaturalization with commercial seeds? Which species are employed? Statements like 'Artificial grasslands mainly include artificial grasslands for controlling degraded grasslands and semi-artificial grasslands for improving and reseeding natural grasslands' does not help to understand this practice.”

Response:

Thank you for your valuable suggestion. We have added the following: ‘Artificial grasslands include areas for growing forage crops for livestock and areas where natural grasslands are improved by reseeding with native and selected commercial species. Their creation aims to boost forage production, control soil erosion in degraded grasslands, and improve the ecological balance of the pasture ecosystem.’

Reviewer #1commented：“L134 Please, add a brief explanation (with references) of what a system dynamic approach is.”

Response

Thank you for your suggestion. A brief explanation has been added in Section 2.2: Application of System Dynamics (SD) Modeling in Decision Support for Grass-Livestock Systems.We have added the following:‘System Dynamics (SD) modeling is one of the common methods in Integrated Assessment Modeling (IAM). It is a simulation and prediction method that can integrate various key elements and form internal feedback. The SD modeling method can provide a favorable tool for the complex grass-livestock (or pasture) systems and the decision-making needs of stakeholders. It is capable of integrating multidisciplinary knowle

---

## [Decision Letter · Decision Letter 1]

5 Jan 2025

Coevolutionary dynamics in the grass-livestock social-ecological system of China's alpine pastoral areas: A case study of the Qilian Mountains region in China

PONE-D-24-30315R1

Dear Dr. Chen,

We’re pleased to inform you that your manuscript has been judged scientifically suitable for publication and will be formally accepted for publication once it meets all outstanding technical requirements.

Kind regards,

Agung Irawan

Academic Editor

PLOS ONE

Additional Editor Comments (optional):

Reviewers' comments:

Reviewer's Responses to Questions

**Comments to the Author**

1. If the authors have adequately addressed your comments raised in a previous round of review and you feel that this manuscript is now acceptable for publication, you may indicate that here to bypass the “Comments to the Author” section, enter your conflict of interest statement in the “Confidential to Editor” section, and submit your "Accept" recommendation.

Reviewer #2: (No Response)

2. Is the manuscript technically sound, and do the data support the conclusions?

Reviewer #2: (No Response)

3. Has the statistical analysis been performed appropriately and rigorously?

Reviewer #2: (No Response)

4. Have the authors made all data underlying the findings in their manuscript fully available?

Reviewer #2: (No Response)

5. Is the manuscript presented in an intelligible fashion and written in standard English?

Reviewer #2: (No Response)

6. Review Comments to the Author

Reviewer #2: The authors made lots of changes according to the suggestions. Therefore, I think it can be accepted now.

7. PLOS authors have the option to publish the peer review history of their article (what does this mean? ). If published, this will include your full peer review and any attached files.

**Do you want your identity to be public for this peer review?** For information about this choice, including consent withdrawal, please see our Privacy Policy .

Reviewer #2: No

---

## [Editor Report · Acceptance letter]

PONE-D-24-30315R1

PLOS ONE

Dear Dr. Chen,

I'm pleased to inform you that your manuscript has been deemed suitable for publication in PLOS ONE. Congratulations! Your manuscript is now being handed over to our production team.

Kind regards,

on behalf of

Dr. Agung Irawan

Academic Editor

PLOS ONE